# Genome-Wide Identification and Expression Analysis of Auxin Response Factor (ARF) Gene Family in *Panax ginseng* Indicates Its Possible Roles in Root Development

**DOI:** 10.3390/plants12233943

**Published:** 2023-11-23

**Authors:** Min Yan, Yan Yan, Ping Wang, Yingping Wang, Xiangmin Piao, Peng Di, Deok-Chun Yang

**Affiliations:** 1State Local Joint Engineering Research Center of Ginseng Breeding and Application, Jilin Agricultural University, Changchun 130118, China; yanmin_1998@126.com (M.Y.); yany@jlau.edu.cn (Y.Y.); wp9859@126.com (P.W.); yingpingw@126.com (Y.W.); pxm52_@163.com (X.P.); 2Graduate School of Biotechnology, College of Life Science, Kyung Hee University, Yongin-si 17104, Gyeonggi-do, Republic of Korea

**Keywords:** *Panax ginseng*, auxin response factor (ARF), root development

## Abstract

Auxin-responsive factors (ARFs) are an important class of transcription factors and are an important component of auxin signaling. This study conducted a genome-wide analysis of the ARF gene family in ginseng and presented its findings. Fifty-three ARF genes specific to ginseng (*PgARF*) were discovered after studying the ginseng genome. The coding sequence (CDS) has a length of 1092–4098 base pairs and codes for a protein sequence of 363–1565 amino acids. Among them, *PgARF32* has the least number of exons (2), and *PgARF16* has the most exons (18). These genes were then distributed into six subgroups based on the results obtained from phylogenetic analysis. In each subgroup, the majority of the *PgARF* genes displayed comparable intron/exon structures. *PgARF* genes are unevenly distributed on 20 chromosomes. Most PgARFs have B3 DNA binding, Auxin_resp, and PB1 domains. The *PgARF* promoter region contains various functional domains such as plant hormones, light signals, and developmental functions. Segmental duplications contribute to the expansion of the *ARF* gene family in ginseng, and the genes have undergone purifying selection during evolution. Transcriptomic results showed that some *PgARFs* had different expression patterns in different parts of ginseng; most *PgARFs* were affected by exogenous hormones, and a few *PgARFs* responded to environmental stress. It is suggested that *PgARF* is involved in the development of ginseng by regulating hormone-mediated genes. *PgARF14*, *PgARF42*, and *PgARF53* are all situated in the nucleus, and both *PgARR14* and *PgARF53* noticeably enhance the growth length of roots in Arabidopsis. Our findings offer a theoretical and practical foundation for exploring *PgARFs*’ role in the growth of ginseng roots.

## 1. Introduction

Transcription factors (TFs) have a vital function in all aspects of plant growth and development, including basal metabolism, cell differentiation and organ formation. These factors are present in all biological processes [1]. TFs serve as crucial nodes in regulatory networks, and variations in their expression levels can influence subsequent pathway gene expression [2,3]. This, in turn, impacts plant morphogenesis as downstream gene activation or inhibition happens [4,5,6]. Transcription factors typically possess multiple modules with distinct structures, serving two fundamental purposes: identifying and binding the specific DNA sequence on the target gene and recruiting additional proteins to facilitate downstream reactions. The DNA-binding domain is responsible for executing these functions, frequently serving as a dimer with a unique dimerization motif, as seen in the DNA-binding domain of the ARF [7,8].

ARF is a transcription factor discovered in the promoter of primary/early growth hormone response genes and specifically binds to the TGTCTC growth hormone response element (AuxRE) [9]. The ARF has three domains: the DNA-binding structural domain (DBD), the middle region (MR), and the phox and Bem1 (PB1) domain [10,11]. Since the three characteristic ARF structural domains mentioned above are functionally independent, certain *ARF* genes without specific protein structural domains are involved in plant growth [12]. The N-terminal DBD of ARF has been identified as a plant-specific B3-type DBD [13]. Phylogenetic trees created with just the DBD protein sequences of ARF closely resemble those generated with the complete protein sequences, indicating that this domain contributes to functional specificity [9,14]. An MR is composed of either an activating domain (AD) or a repressor domain (RD). The function of the MR to either activate or inhibit is determined by its amino acid composition [15]. PB1 resembles AUX/IAA structural domains III and IV. It often forms homodimers with ARF proteins or heterodimers with AUX/IAA proteins [16,17].

Several *ARFs* are believed to function in an auxin-dependent manner, and the pathway that relies on auxin-dependent signaling is one of the most fully described models. The pathway operates via the disinhibition mechanism, whereby auxin prompts the breakdown of AUX/IAA proteins, thereby releasing dimerized ARF proteins and activating downstream genes [18]. Other ARFs are thought to have alternative and auxin-independent mechanisms of action, such as the AtARF2-MR-TPL mutual inhibition that underlies root hair formation [19]. ARFs are also incorporated into more complex transcription factor complexes to integrate signals from additional hormone signaling pathways and react to environmental stresses [20].

Up to now, the *ARF* genes of *Arabidopsis* (23 *ARFs*), rice (25 *ARFs*), soybeans (51 *ARFs*), peaches (17 *ARFs*), carnations (29 *ARFs*), peanuts (30 *ARFs*), and eggplants (20 *ARFs*) have been identified through whole-genome analysis [16,21,22,23,24,25,26]. There are 23 *ARF* genes in Arabidopsis, of which 1 is a pseudogene. *ARF1* is the first transcription factor identified by a yeast one-hybrid assay containing a core motif [27]. *ARF1* is a transcriptional repressor [28]. *ARF2* is associated with senescence and abscission of plant flowers and leaves. *ARF1/ARF2* double mutants had similar phenotypes to single mutants, but both phenotypic traits are further enhanced, indicating that *ARF1* and *ARF2* act partially redundantly [28,29]. *ARF3* and *ARF4* contribute to the development of reproductive and nutritional tissues [30]. *ARF5* is involved in both the formation of vascular tissue and hypocotyl development [31,32]. Functional redundancy exists between *ARF6* and *ARF8*, which are associated with pistil and stamen reproductive capacity. *ARF6/ARF8* double mutants display shortened stamen filaments and delayed anther dehiscence, leading to the manifestation of female sterility in plants [33]. The double mutation of *AtARF7/AtARF19* hindered the formation of adventitious roots and resulted in a decrease in lateral root quantity. However, a single mutation did not significantly affect the development of adventitious and lateral roots, suggesting functional redundancy between the two [34]. There were 25 *OsARFs* identified from rice, and in the prediction of OsARF protein, nine were identified as activators and the rest as inhibitors. *OsARF1* and *OsARF23* actively respond to auxin with increased transcripts, in contrast to *OsARF5*, *OsARF14*, and *OsARF21* with decreased transcripts [35]. *SlARF2* of *Solanum lycopersicum* regulates lateral root growth and flower development [36]. In the peach, *PpARF7* and *PpARF12* are involved in flower bud formation and fruit softening, respectively [23].

While the ARF gene family plays an essential part in the majority of plant activities, spanning from germination to senescence, as well as influencing plant responses towards exogenous hormones and abiotic stresses, no research has yet identified and analyzed *ARF* genes in *Panax ginseng*. As a result, we conducted a genome-wide identification and analysis of the ginseng *ARF* gene family, constructing a phylogenetic tree and categorizing the *ARF* genes into distinct branches using a phylogenetic approach. The basic composition of the *PgARF* gene was analyzed, and the similarities and differences in structure and composition of the *PgARF* gene and protein were obtained. The expression data of *PgARF* genes were analyzed under environmental stress conditions and in different plant parts. Additionally, exogenous hormones were applied to ginseng, and its transcriptome data were analyzed to determine the expression variables of *PgARF* genes. The results showed the expression of different *PgARF* genes under different exogenous hormone treatments, indicating that the expression patterns of different *PgARF* genes differ. The roles of *PgARF14*, *PgARF42*, and *PgARF53*, which may affect ginseng root formation, were investigated in *Arabidopsis thaliana*. Their roles in root growth were preliminarily probed to provide a theoretical and experimental framework for the next step of research on the molecular mechanisms by which phytohormones and *PgARF* genes play a role in the growth and development of ginseng roots.

## 2. Results

### 2.1. Identification, Classification, and Phylogenetic Analysis of PgARF

We identified 53 genes encoding *ARF* transcription factors in the ginseng genome database and named them *PgARF1*-*PgARF53* (Appendix A). The complete *PgARF* sequences are listed in Appendix A. The open reading frame of the *PgARFs* ranges from 1092 (*PgARF53*) to 4698 (*PgARF46*) bp, which predicts the encoding of proteins with 363–1565 amino acids. The molecular weight of the protein varies between 40,550.85 and 173,376.03 Da. Among them, *PgARF32* has the least number of exons (2), and *PgARF16* has the most exons (18). The isoelectric point range is 5.23 (*PgARF50*)-9.41 (*PgARF48*) pH (Appendix A).

We generated a phylogenetic tree by employing the complete amino acid sequences of 53 PgARFs, 22 AtARFs, and 27 OsARFs. The phylogenetic analysis revealed that the ARF family proteins of *Panax ginseng*, *Oryza sativa*, and *Arabidopsis thaliana* can be classified into six distinct groups. There are subgroups Ia, Ib, and Ic in group I; subgroups IIa and IIb in group II; and group III (Figure 1). Subgroup Ia contains six ARF proteins (four PgARFs, one AtARF, one OsARF); subgroup Ib contains 14 ARF proteins (8 PgARFs, 2 AtARFs, 4 OsARFs); and the Ic subgroup contains 10 ARF proteins (4 PgARFs, 2 AtARFs, 4 OsARFs). Subgroup IIa contains 32 ARF proteins (14 PgARFs, 12 AtARFs, 6 OsARFs), and subgroup IIb contains 13 ARF proteins (7 PgARFs, 2 AtARFs, 4 OsARFs). Group III includes 27 ARF proteins (16 PgARFs, 3 AtARFs, 8 OsARFs). This classification result is consistent with that of physic nut [37]. The analysis of phylogenetic tree results (Figure 1) also showed that most PgARFs were more closely related to AtARFs than OsARFs. For example, in the two main branches in subgroup Ia, four PgARFs (PgARF4, PgARF42, PgARF50, PgARF52) are in the same branch as AtARF5, and the other branch is only OsARF11.

We have examined the exon distribution of the *PgARFs*. The results showed that all *PgARFs* had multiple exons (Figure 2B). Most *PgARF* members within the same group demonstrated comparable structures in their exons and introns. For example, six *PgARFs* of subgroup Ib have 14–15 exons; 12 *PgARFs* of group III have 3–4 exons. Although the number of exons of *PgARF* varies amongst subgroups, they are largely similar. The exons–introns of the *PgARFs* in each subgroup had similar structures in number and length. This result further supports the classification of *PgARF* family members in this study.

### 2.2. Protein Conservation Motif Analysis of PgARF

Domain features of 53 PgARF proteins were analyzed using the MEME online analysis tool. Ten individual motifs were found in 53 PgARF proteins, which we named motifs 1–10 (Figure 2A) (Appendix A). The DBD domain sequence contains motif1, motif3, and motif10; the Auxin_resp domain corresponds to motif8; and the PBI domain has motif9 and motif5. It is worth mentioning that motif5 also corresponds to the protein sequence of the AUX_IAA domain. These motifs detected homologous sequences in *Arabidopsis* and rice ARF proteins [21], demonstrating that their composition is highly conserved across species. However, motif2, motif4, motif6, and motif7 lack homologous sequences in the protein databases SMART and Pfam, so the function of some putative motifs is unclear.

The findings indicate that each PgARF protein possesses a consistent B3 domain at the N-terminus (Figure 2A). Most PgARF proteins contain the Auxin_resp domain, except PgARF26 and PgARF35 (Figure 2A). The PB1 domain, also known as CTD, mediates protein–protein interactions and can control the expression of auxin-responsive genes through homologous or heterologous dimerization of ARF-Aux/IAA or ARF-ARF. In this study, 11 PgARF proteins do not contain CTD, namely PgARF2, PgARF6, PgARF9, PgARF15, PgARF20, PgARF32, PgARF39, PgARF46, PgARF48, PgARF52, and PgARF53, and the proportion of PgARFs (20.75%) with a CTD truncation was much lower than that of MtARFs (54%). CTD-truncated MtARFs in *Medicago truncatula* provide evidence for the auxin-independent regulation of *ARF* genes [38]. Therefore, it is speculated that these CTD-truncated PgARFs may also regulate plant life activities in an auxin-independent pathway. Furthermore, 32 PgARFs have AUX_IAA domains, which may affect their function. It is worth mentioning that among the branches of the phylogenetic tree, the motifs contained in each branch are structurally specific, which means that *PgARFs* in the same group may have similar functions.

### 2.3. Analysis of Cis-Elements of PgARF Family Promoters

The cis-acting elements that can regulate gene expression contained in the promoter (upstream 2000 bp) region of *PgARFs* were analyzed using the PlantCare online tool (Figure 3). Cis-acting elements are divided into seven types: developmental-related elements, environmental stress-related elements, hormone-responsive elements, promoter-related elements, binding site-related elements, light-responsive elements, and other elements. To investigate the impact of abiotic stress on the expression of *PgARF*, this study focused on the analysis of hormone-responsive factors and environmental stress-related factors (Figure 3).

The hormone-responsive elements in the *PgARF* family account for about 3.21% of the total elements, including elements related to auxin (TGA-element, TGA-box, AuxRR-core, AuxRE), gibberellin (TATC-box, P-box, GARE-motif)), abscisic acid (ABRE), salicylic acid (TCA-element), and methyl jasmonate (TGACG-motif, CGTCA-motif); these hormones may affect the expression level of *PgARF*. The environmental stress-related elements include drought (MBS) and low temperature (LTR), and these two environmental stresses may affect genes with these two elements.

### 2.4. Chromosomal Distribution and Collinearity Analysis of PgARF

The analysis of chromosome distribution revealed that 53 *PgARFs* exhibited uneven distribution across 20 chromosomes (Figure 4) (Appendix A). Among them, there are the most genes distributed on chr4 and chr10, with five *PgARFs* each; then, there are chr3, chr5, and chr12 with four *PgARFs*; there are three *PgARFs* on chr2, chr6, chr19, and chr22, respectively; and only one *PgARF* was distributed on chr8, chr18, and chr20; The remaining eight chromosomes contain two *PgARFs* (Figure 4). 

According to previous reports [22], 25 segmental duplications were identified in *PgARFs* (Figure 4, Appendix A). To investigate the possible selective pressure on *PgARF*, Ka/Ks ratios were calculated. The findings indicate that *PgARF* is predominantly impacted by purification selection during its evolution, as all *PgARF* gene pairs had Ka/Ks values less than 1 (Appendix A).

### 2.5. Synteny and Evolutionary Analyses of PgARF and Other Plants’ ARF

To explore the potential evolutionary clues of *PgARF*, a comparative collinear map of *Panax ginseng* and the other two species of Panax was constructed (Figure 5). The two plants are *Panax quinquefolius* and *Panax notoginseng*, respectively. According to the comparative collinear map of the three plants (Figure 5), 33 *PgARFs* were homologous to the *ARF* genes of *Panax quinquefolius*, and 32 *PgARFs* were homologous to the *ARF* genes of *Panax notoginseng*. In the report by Wang [39], the Modern Panax genome is formed by the core-eudicot genome undergoing three whole-genome duplications (WGD); these three WGDs include γ-triplication events and Panax-specific WGDs (Pg-β and Pg-α). The difference in the number of homologous genes between *PgARF*-*PnARF* and *PgARF*-*PqARF* is tiny, confirming the previous conclusions. 

### 2.6. Expression Profiles of PgARF in Different Tissues and Growth Years

As depicted in Figure 6A, around 24 *PgARFs* (45.28%) are expressed in the roots, stems, and leaves of ginseng aged 1–5 years, with some genes showing tissue-specific expression. Compared with other *PgARFs*, the expression levels of *PgARF17*, *PgARF47*, *PgARF14*, *PgARF28*, *PgARF30*, and *PgARF28* were relatively high. Of all the genes, *PgARF17* stands out as having a stable and high level of expression in the root, suggesting that its presence may be critical for the development of root tissue. The expression of *PgARF14* was remarkably high in the stem, indicating its significant involvement in stem growth. *PgARF30* expression in leaves was significantly higher than in roots and stems, being the highest of the three. This gene is potentially essential for leaf growth and development. *PgARF28* displays high expression levels in 2-year-old stems and leaves, indicating a possible role in the development of ginseng stems and leaves (Figure 6A).

Using existing transcriptome data [40], we analyzed the expression of *PgARFs* in different parts of ginseng, including the fiber root, leg root, main root epiderm, main root cortex, rhizome, arm root, stem, leaf peduncle, leaflet pedicel, fruit peduncle, fruit pedicel, fruit flesh, and seed. A heatmap depicting the expression levels in these various parts of ginseng roots (Figure 6B) was generated. Gene expression results showed that about 37 *PgARFs* (69.81%) were expressed in all sites. *PgARFs* exhibit distinct expression patterns in diverse tissues. For example, *PgARF17* is the most expressed gene in all roots, leaves, fruits, and seeds. Other sections exhibit substantial expression, suggesting that this gene plays a crucial function in the complete process of plant growth and development. *PgARF14* was significantly expressed in branch roots, fibrous roots, and rhizomes. The expression level of *PgARF30* in leaves was slightly lower than *PgARF17*, and the expression level in leaves was much higher than that in petioles and pedicels. *PgARF28* was the most expressed gene in the stem, but the expression level of this gene in each tissue was not specific. *PgARF7* was the most expressed gene in fresh fruit, but its expression was also significant in fruit stems (Figure 6B).

### 2.7. Expression Profiles of PgARF under Different Environmental Stresses and Hormone Treatments

We analyzed the response of PgARF under abiotic stress treatments using the data of cold, heat, drought, and salt from the publicly available ginseng dataset [41]. As shown in Figure 7A, many *PgARFs* showed similar expression trends after abiotic stress treatment. The results showed that three *PgARFs* were not expressed in any tissue under any treatment, namely *PgARF9*, *PgARF33*, and *PgARF39*. The *PgARFs* under different treatments had different expression patterns: In 1-year-old leaves, only *PgARF53* was significantly changed after a 1-week heat treatment compared with the control group. However, seven *PgARFs* (*PgARF5*, *PgARF23*, *PgARF32*, *PgARF43*, *PgARF46*, *PgARF51*, *PgARF53*) were significantly changed during the 3-week heat treatment. The expression levels of *PgARF53* underwent significant changes during the 1-week and 3-week heat treatments. This indicates a potential role for the protein in the response of ginseng to heat stress.

In the cold treatment, notable alterations occurred in the expression of three *PgARFs* (*PgARF15*, *PgARF50*, *PgARF51*), with the expression of *PgARF25* being altered by over 4-fold, potentially implicating *PgARF25* in contributing to ginseng’s drought resistance. It is worth mentioning that the expression of *PgARF51* changes when ginseng is subjected to cold and heat stress, and *PgARF51* may regulate the response of ginseng to external temperature. 

Under drought stress, expression levels of *PgARF2*, *PgARF42*, and *PgARF46* significantly increased compared to the control group, while the expression of *PgARF52* was more than four times higher. In salt treatment, only *PgARF2* and *PgARF32* showed significant increases compared to the control group.

To examine the impact of external hormones on *PgARF* expression levels, four exogenous hormones were administered to ginseng (auxin, gibberellin, cytokinin, and abscisic acid). In analyzing them at the transcriptomic level, the expression heat map of *PgARFs* under different hormone stress was obtained (Figure 7B). Compared to the blank control group, the 6BA hormone treatment resulted in minimal changes in the expression of *PgARFs*, with only two genes, *PgARF4* and *PgARF42*, exhibiting significant changes. Under ABA stress, nine *PgARFs* (16.98%) exhibited significant changes in expression levels, specifically *PgARF4*, *PgARF18*, *PgARF32*, *PgARF34*, *PgARF42*, *PgARF46*, *PgARF48*, *PgARF50*, and *PgARF51*. In the group of subjects treated with IAA, 10 *PgARFs* (18.87%) exhibited noteworthy variations in their levels of expression when contrasted with the control group that received no treatment. Seven of these *PgARFs* exhibited notable changes, including *PgARF7*, *PgARF11*, *PgARF17*, *PgARF27*, *PgARF28*, *PgARF42*, and *PgARF53*. Meanwhile, the expression levels of *PgARF4*, *PgARF40*, and *PgARF50* were significantly elevated. Under GA stress, both the increased expression of *PgARFs* and the significantly increased number of *PgARFs* were the highest. Compared with the blank control, the expression levels of 15 *PgARFs* (28.30%) were significantly increased in GA treatment, and the expression levels of 10 *PgARFs* were significantly increased under GA stress (respectively, *PgARF10*, *PgARF14*, *PgARF21*, *PgARF22*, *PgARF26*, *PgARF36*, *PgARF47*, *PgARF49*, *PgARF51*, *PgARF52*). Moreover, the expression levels of five *PgARFs* increased significantly, namely *PgARF1*, *PgARF25*, *PgARF31*, *PgARF42*, and *PgARF46*. Among them, this experiment showed that *PgARF31* had the most increased expression among all hormone treatments. This result indicates that *PgARFs* may be insensitive to 6BA and more involved in plant life activities mediated by IAA and GA. 

The *ARF* genes in *Arabidopsis* have been classified into three classes, with class A containing the *ARF* genes *AtARF5*, *AtARF6*, *AtARF7*, *AtARF8*, and *AtARF19*. Class A *ARFs* are thought to work in a growth factor-dependent manner, but class A *ARFs* can also respond to environmental signals through auxin-independent pathways and regulate plant growth and development. In addition, A class ARF-AUX/IAA interactions are relevant in different developmental environments. However, this does not mean that ARFs of class A interact with other molecular signals only through the PB1 structural domain; for example, interactions with chromatin remodelers SPLAYED (SYD) and BRAHMA (BRM) have been described only in *AtARF5*, which are involved in ARF5-MR, but the specific regions that regulate them have not been identified. Similarly, in the context of auxin control of lateral root formation, interactions with the Mediator complex are only available in *AtARF7* and *AtARF19*, but the structural domains with which they interact are not yet known [17].

In the phylogenetic tree of this study (Figure 1), *PgARF14*, *PgARF42*, and *PgARF53* are all in the branch I, but clustered with *AtARFs* of different A classes: *PgARF14* and *AtARF6* and *AtARF8* are in subgroup Ib, *PgARF42* and *AtARF5* are in subgroup Ia, *PgARF53* and *AtARF7* and *AtARF19* are in subgroup Ic. In terms of the structural domains of the protein molecules, both PgARF14 and PgARF42 contained DBD, Auxin_resp, and PB1, and PgARF53 had only DBD, Auxin_resp, and no PB1. However, in exogenous hormone treatments, *PgARF14* showed a significant change in expression after GA treatment, and *PgARF42* responded to 6BA and ABA. On the contrary, *PgARF53*, which has no PB1 structure, did show a significant increase in expression after IAA and GA treatments. Therefore, *PgARF14, PgARF42*, and *PgARF53* were selected for further experiments.

### 2.8. Subcellular Localization of PgARF14, PgARF42, and PgARF53

The expression vectors, composed of the target genes (35S:GFP-*PgARF14*, 35S:GFP-*PgARF42*, and 35S:GFP-*PgARF53*), were transformed into *Arabidopsis* protoplasts along with empty vectors. The obtained subcellular localization results (Figure 8) confirmed the nuclear expression of *PgARF14*, *PgARF42*, and *PgARF53*, as clear GFP fluorescent signals were detected in the nucleus.

### 2.9. PgARF14 and PgARF53 Increase the Root Length in Arabidopsis

Root lengths of *Arabidopsis* seedlings were evaluated utilizing Image J 1.53t. A one-way ANOVA was conducted in GraphPad for data analysis. Transgenic *Arabidopsis* root length data were collected for T2 generations of *PgARF14*, *PgARF42*, and *PgARF53*. According to the results of the analyses (Figure 9), the root length data of *PgARF14* were highly significantly different from those of WT-*PgARF14*. The root length of *PgARF42* was not statistically different from WT-*PgARF42*. The root length data of *PgARF53* was significantly different from WT-*PgARF53*. The root lengths of transgenic *Arabidopsis* of *PgARF14* and *PgARF53* were longer than their wild-type controls.

## 3. Discussion

Currently, there exists limited knowledge regarding ginseng’s growth, development, and response to abiotic stress. And the homology and function of *PgARF* are not clear. A total of 53 *PgARFs* were identified in this study. In the ginseng genome, the number of *PgARF* family genes exceeds that of Arabidopsis, which has 23 *AtARFs*, or rice, which has 25 *AtARFs* [21,42]. The number of genes depends on the genome size and gene duplication events of the plant [43], so one possible explanation for the higher number of *PgARFs* is that segmental duplications play an important role in the expansion of the *PgARF* gene family. We divided PgARFs into six subgroups, similar to those in rice and *Arabidopsis* (Figure 1). In the phylogenetic tree diagram, PgARFs and AtARFs are significantly more closely related. In addition, the phylogenetic tree also showed that there were 16 PgARFs (30.2%, 16/53) in group III, while only three AtARFs (13.6%, 3/22) were classified into this group (Figure 1). This implies that this gene set may have been present in both *Arabidopsis* and ginseng. However, it was lost from the *Arabidopsis* lineage during evolution [23].

Evolutionary relationships between species can also be revealed by the degree of difference in intron–exon patterns of genes [23]. The intron–exon splicing arrangement and intron number of *PgARFs* in the ginseng genome are similar to those reported in Arabidopsis, rice [21], and maize [35]. For example, the ginseng *ARF* gene family has the least number of introns in the group III genes (Figure 2.). Similarly, in Arabidopsis, rice and maize, there is also a subgroup of genes with only 1–3 introns [21,35]. Analysis of the conserved domains of PgARF proteins showed that all PgARF proteins have a conserved N-terminal DBD domain (motif 1), indicating that *ARFs* are highly conserved in various plants during evolution. Furthermore, each subgroup has a specific motif composition structure (Figure 2). These features of conserved domains of PgARF proteins are also found in plants such as Arabidopsis, rice, and maize [35]. Overall, most *PgARF* genes in the same group have similar gene structures and conserved motifs, which further supports their classification and evolutionary relationship between groups described in this study [37].

The level of gene expression is frequently associated with its function. Thus far, detailed investigations into the expression of *ARF* genes in ginseng have not been conducted despite the expression patterns of *ARF* genes in other plants, including rice, maize, and Arabidopsis, having been established [20,34]. In this study, the expression of *PgARF* under different hormone stresses was detected at the transcriptional level, and the expression of *PgARFs* in different groups at different times and under different stresses was also collected. Studies have shown that many *ARFs* regulate the development of plant roots, stems, leaves, and flowers [12,44]. The analysis showed that the expression of *PgARF17* was the highest in the root (Figure 6B), but it was significantly expressed in other parts, and the expression level of *PgARF17* in the root was also the highest in the expression heat map in different years (Figure 6A), although *PgARF17* was expressed in other parts of ginseng. The expression of *PgARF17* is not specific, but in the expression heatmap of different years, there is a high level of expression in the root in each year, and in the phylogenetic tree clustering (Figure 1), *PgARF17* and *AtARF2* are in the same subgroup IIa. Hence, it is hypothesized that *PgARF17* is a crucial *ARF* gene in the growth of ginseng roots. The expression level of *PgARF30* in leaves is slightly lower than that of *PgARF17*, but its expression level in leaves is much higher than in petioles and pedicels. It has been reported to have partial functional redundancy with its homologous gene *AtARF2* in Arabidopsis, which regulates leaf senescence and flower organ shedding [28]. Based on the specific high-level transcription of *PgARF30* in leaves, it is inferred that it could contribute to the growth and development of leaves. While *AtARF5* is necessary for radicle formation, the expression levels of the four genes belonging to the same group *PgARFs* are all low (Figure 6B). It is speculated that the regulatory effect of *ARFs* on the growth and development of ginseng roots may be different from that of Arabidopsis.

Phytohormones are involved in various plant responses to environmental stimuli and stresses by altering the expression levels of many *ARF* genes. However, few studies have investigated ginseng’s response to abiotic stress and related signal transduction. In this study, we built expression heat maps of *PgARF* family genes in response to several hormones. *ARF* has an auxin-dependent signal transduction pathway and an auxin-independent signal transduction pathway when regulated by auxin [40]. Under auxin stress, a total of 10 *PgARFs* were significantly increased in expression, of which 7 *PgARFs* (*PgARF4*, *PgARF7*, *PgARF11*, *PgARF17*, *PgARF40*, *PgARF42*, *PgARF50*) had AUX/IAA domains. *PgARF4* and *PgARF50* clustered with *AtARF5* (Figure 1), and these two genes may be involved in developing ginseng flowers. Among auxin-independent signaling, one is related to root hair development, and inhibition through the AtARF2-MR-TPL interaction underlies root hair formation. AtARF2-MR comprises both an EAR motif and a BRD domain, which facilitates its binding to both the N-terminal and the C-terminal sections of TPL/TPR [19]. Single mutations of these two inhibitory domains had less effect on *AtARF2* inhibition in root hairs, whereas double mutations strongly affected this inhibition. The two mutations additionally impacted the time of flowering and size of seeds, which are two phenotypes distinctive to *ARF2* mutants. *PgARF28* is sensitive to exogenous auxin and has no AUX/IAA structure, and *PgARF28* is significantly expressed in roots (Figure 9), so it is speculated that *PgARF28* may play a role in the development of ginseng roots. Among the 15 *PgARFs* whose expression levels were significantly increased under GA3 stress, 7 *PgARFs* (*PgARF1*, *PgARF10*, *PgARF14*, *PgARF21*, *PgARF26*, *PgARF31*, and *PgARF47*) were in the Ib subfamily—the subfamily where *AtARF6* and *AtARF8* belong.

Transcriptomic analysis has demonstrated that the transcriptional levels of numerous *ARF* genes alter under plant responses to abiotic stressors. Auxins play critical roles in plant responses to abiotic stresses through complex metabolic and signaling networks [45]. In bananas, most *ARF* genes are altered in expression under salinity and osmotic stress [46]. In rice, *OsARF11* and *OsARF15* display differential expression under salt stress conditions, indicating their role in the plant’s response to stress [47]. This study revealed that a multitude of *PgARFs* are capable of responding to abiotic stress factors such as salt, cold, heat, and hormonal stress at the transcriptional level. Moreover, it was observed that the majority of these genes fall under the IIa and IIb subgroups (Figure 1 and Figure 7). The intron–exon structure of the group II gene sequence is different from that of other groups, which may affect its function and action pathway.

The experimental findings reveal that transforming the target gene of *Arabidopsis thaliana* resulted in a significant difference in the root length of *PgARF14* when compared to the WT-*PgARF14*. Furthermore, in the results of the phylogenetic analysis of this study (Figure 1), *PgARF14* was categorized in subgroup Ib of the phylogenetic tree, grouped with *AtARF8*, which has been shown to regulate not only hypocotyl development but also to influence the growth of the primary root [48]. *PgARF14* contains cis-acting elements which react specifically to gibberellins. Treatment with exogenous gibberellins may result in a significant increase in its expression. The role of auxin in activating GA biosynthesis is established, but the intricate levels of interaction between these two hormones are still being investigated [49]. In addition, gibberellic acid (GA) has the potential to affect the regulatory pathway of auxin and cytokinin, specifically ARR1-SHY2-PIN. Elevated concentrations of GA impede *ARR1* expression by triggering the breakdown of Della proteins, which takes place during the growth stage of meristematic tissue. Consequently, this results in low levels of *SHY2* that promote cytokinesis [49,50]. The DELLA-ABI4-HY5 module, in turn, can act as a molecular chain that integrates GA and light signaling to control hypocotyl elongation, with DELLAs directly interacting with and inhibiting the DNA-binding activity of ABI4. In turn, ABI4 binds to ELONGATED HYPOCOTYL 5 (HY5), a key factor in light signaling, and its feedback regulates the *GA2ox* GA catabolism gene expression, thereby regulating GA levels [5]. Therefore, despite being expressed at high levels in rhizomes and fibrous roots rather than the taproot, *PgARF14* is thought to regulate the growth and development of the taproot.

There was no significant variance observed in the root length between WT-*PgARF42* and *PgARF42*. According to the phylogenetic tree, *PgARF42* was categorized in subgroup Ia along with *AtARF5*. *AtARF5* has been reported to have a vital role in initiating and localizing lateral roots. However, this function can only be observed at the cellular level [51]. Therefore, other methods may be required to observe potential differences between *PgARF42* transgenic plants and wild-type controls. *PgARF42* showed sensitivity to drought stress. It has been shown that MiRNA390, ta-siRNA, and ARF co-regulate the auxin pathway, e.g., the promotion of lateral root growth in poplar under salt stress [52]. It is therefore hypothesized that *PgARR42* may affect the growth and development of ginseng in other ways.

The root length of *PgARF53* differed significantly from WT-*PgARF53*. *PgARF53* is situated in the Ic subgroup of the phylogenetic tree, and *AtARF7* and *AtARF19* are also clustered in this group. *AtARF7* and *AtARF19* are key regulators of lateral root growth and development [53]. *PgARF53* expression was significantly elevated after IAA and GA treatments, so it may regulate ginseng growth and development through the molecular signaling network between GA and IAA. The expression of *PgARF53* in *Arabidopsis* resulted in root length variations, likely due to the use of the 35S promoter, which activates *PgARF53* expression throughout all parts of Arabidopsis. However, the heatmap (Figure 6) showed that the expression of PgARF53 in ginseng may be tissue-specific and lower in ginseng roots. In the phylogenetic tree (Figure 1), none of the other PgARF genes (*PgARF22*, *PgARF36*, *PgARF43*) clustered in the same subgroup with *PgARF53* were tissue-specific in expression, and they were widely expressed in ginseng at different ages and tissue sites, whereas *PgARF53* was structurally different from the other three homologous *PgARF* genes: PgARF53 lacks the PB1 structural domain. Therefore, although *PgARF53* can cause root elongation in Arabidopsis, its role in ginseng development needs to be verified by further studies.

## 4. Materials and Methods

### 4.1. Identification and Annotation of ARF Genes in Panax ginseng

The *Panax ginseng* genomic data were downloaded from the National Center for Biotechnology Information (https://www.ncbi.nlm.nih.gov, (accessed on 3 April 2022)) (https://ngdc.cncb.ac.cn/, (accessed on 3 April 2022)) [39], under the project number PRJNA752920. The *ARF* datasets for *Arabidopsis thaliana* and *Oryza sativa* are from PlantTFDB (http://planttfdb.cbi.pku.edu.cn/, (accessed on 3 April 2022)).

The Hidden Markov Model (HMM) was used to identify PgARF in the ginseng genome. In HMM profiles, the parameters were default, and the cutoff was 0.01. Then, the candidate PgARFs containing the ARF signature domains were detected and screened through the Pfam (http://pfam.xfam.org/, (accessed on 10 April 2022)) and SMART (http://smart.embl.de/, (accessed on 12 April 2022)) databases. Candidate PgARFs were further validated using the NCBI Conserved Domain Database (CDD) (http://www.omicsclass.com/article/310, (accessed on 15 April 2022)) to ensure that they contained ARF domains.

The physicochemical properties of PgARF proteins, including length of DNA and protein sequences (len_cds/len_pro), molecular weight of protein (MW), and isoelectric point (pI), were analyzed using online tools—the Sequence Manipulation Suite (http://www.detaibio.com/sms2/index.html, (accessed on 22 April 2022)) [54].

### 4.2. Phylogenetic analysis of PgARF with AtARFs and OsARFs

Full-length ARF proteins were aligned using ClustalW with default parameters. The phylogenetic tree of *Panax ginseng* ARF was established by the Maximum Likelihood of IQ-TREE [55]. For the ARF phylogenetic tree between *Panax ginseng* and other species, MEGAX was used to make the tree using the neighbor-joining method and 1000 bootstrap replicates. Phylogenetic tree results were further annotated in the online tool Evolview (https://www.evolgenius.info/evolview/#/treeview, (accessed on 10 May 2022)). [56]

### 4.3. Exon–Intron Structure, Conserved Motif, and Cis-Acting Element Analysis

Conserved motifs of ginseng ARF were analyzed using the MEME Suite (MEME—Submission form (meme-suite.org) (accessed on 10 May 2022)) [57], and the CDS structure was visualized with TBtools [58]. The cis-acting elements of ginseng *ARFs* were analyzed online on the PlantCARE website (https://bioinformatics.psb.ugent.be/webtools/plantcare/html/, (accessed on 10 May 2022)) [59], and the results were annotated using Tbtools.

### 4.4. Chromosomal Location, Duplication, Synteny, and Evolution Analyses PgARF

The MCScanX software (https://github.com/wyp1125/MCScanX, (accessed on 18 March 2019)) [60] facilitated interspecies and intraspecies covariance analysis of proteins. Duplicate gene classifier scripts within the program quantified various forms of duplication, including WGD, segment, tandem, dispersed, and proximal duplications. The resulting data were visualized using Circos [61]. Non-synonymous substitution rates (Ka) and synonymous substitution rates (Ks) were calculated for duplicate gene pairs using KaKs-Calculator-2.0. Ka/Ks ratios were used to analyze environmental selection pressures [62].

### 4.5. PgARF Gene Expression Analysis

The RNASeq datasets of 14 different tissues of ginseng are from NCBI (PRJNA302556) [40]. Downloaded 15 RNA-Seq datasets of ginseng under abiotic stress from the Ginseng Genome Database (http://ginsengdb.snu.ac.kr/transcriptome, (accessed on 20 September 2022). php, No.24-No.38) [41]. The datasets of different years of ginseng are from NCBI (PRJNA762437) [63]. The clean reads were aligned with the ginseng genome data using HISAT 2.2.1 software. The Cufflinks were merged with Cuffmerge and the expression value of each transcript was analyzed with Cuffdiff. Differentially expressed genes (DEGs) between different samples were identified using the fragments per kilobase of exon per million mapped reads (FPKM) method [64].

### 4.6. Subcellular Localization

Vectors were constructed using 35S-GFP, *PgARF14*, *PgARF42*, and *PgARF53* to generate 35S:GFP-*PgARF14*, 35S:GFP-*PgARF42*, and 35S:GFP-*PgARF53*, respectively. Protoplasts were then prepared from cells taken from the 5th to 7th leaves of *Arabidopsis* plants aged 3–4 weeks. The protoplasts that were transformed using 35S:GFP-*PgARF14*, 35S: GFP-*PgARF42*, and 35S: GFP-*PgARF53* were subjected to observation and photography under confocal microscopy.

### 4.7. Plant Materials

Ginseng seedlings were grown in the Plant Physiology Laboratory of Jilin Agricultural University, China. Ginseng seeds were vernalized and planted in sandy soil. Cultivation was then carried out in a greenhouse at 25 °C. The photoperiod was 16 h light/8 h dark. Five-week-old ginseng seedlings with three real leaves and growing well were selected. Ordinary distilled water was used as a blank control group. Distilled water, ABA (50 mM), IAA (10 mM), 6BA (75 mM) and GA3 (100 mM) were sprayed evenly on the leaves of ginseng seedlings, respectively. There were five groups of ginseng seedlings with four hormones and a blank control, with three biological replicates in each group. Leaf samples collected at 5 h after hormone treatment were cryopreserved in a −80 °C ultra-low temperature freezer until analysis.

*Arabidopsis* (wild-type Col-0) was used for this study. The coding sequences of *PgARF14*, *PgARF42,* and *PgARF53* were inserted into the pCambia1300-3×FLAG vector. The resulting recombinant vector was transformed into *Arabidopsis* plants via inflorescence infiltration. PCR identification was performed on positive plants screened using 1/2MS medium containing hygromycin, and the electrophoresis map of the PCR identification results is shown in Appendix A. Positive plants were selected for further cultivation and phenotype observation. *Arabidopsis* chamber growth conditions were temperature 21 °C, illuminance 2000–3000 lx, light duration 14 h/day, humidity 40–60%, nutrient soil/vermiculite/perlite = 1:1:1. a light density of ~120 µmol m^−2^ s^−1^.

### 4.8. Statistical Analysis

The lengths of the roots of *Arabidopsis* were measured via ImageJ 1.53t. Subsequently, a one-way ANOVA was carried out for the root length data with GraphPad. The statistical significance was determined based on probability values, where *p* > 0.05 indicated no significant difference, 0.01 < *p* < 0.05 indicated a significant difference, and *p* < 0.01 indicated a highly significant difference.

## 5. Conclusions

In this study, we identified 53 *PgARF* genes in the ginseng genome. Phylogeny, gene structure, and functional motifs were also analyzed. *PgARF* genes were clustered into six different subgroups, similar to previous phylogenetic studies of *Arabidopsis ARF*. Analysis of the transcriptome data indicated that *PgARF* had different expression patterns in different parts of ginseng; most *PgARFs* were affected by exogenous hormones, and a few *PgARFs* were responsive to environmental stresses. This suggests that *PgARF* can potentially participate in various stages of ginseng growth and regulate a variety of hormones to participate in ginseng growth and development. Overexpression of both *PgARF14* and *PgARF53* regulated the elongation of *Arabidopsis* roots, whereas the effect of *PgARF42* on the length of root was not significant. The expression of *PgARF53* in ginseng may be tissue-specific. Moreover, it lacks the PB1 structural domain, which makes the mechanism of its expression in ginseng potentially different from that of *PgARF14*. Our study provided us with an understanding of the diversity of *ARF* genes in ginseng and provided a theoretical and experimental basis for further studies on the effects of hormones on ginseng growth and development. In addition, deeper studies on *PgARF* are needed to explore the molecular mechanisms by which it functions in the hormone signaling network to regulate ginseng growth and development.

## Figures and Tables

**Figure 1 plants-12-03943-f001:**
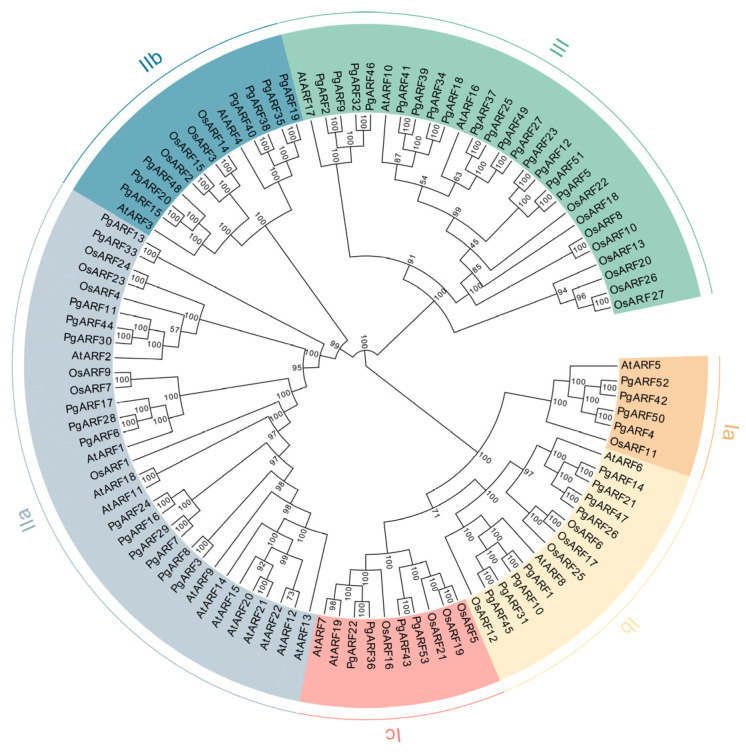
Phylogenetic tree of PgARF proteins. Phylogenetic tree based on the ARFs protein family in three plants. The phylogenetic tree has three branches. Clade I is subdivided into Ia, Ib, and Ic, and clade II into IIa and IIb. At, Arabidopsis; Os, rice; Pg, ginseng. The number stands for the confidence of the branch.

**Figure 2 plants-12-03943-f002:**
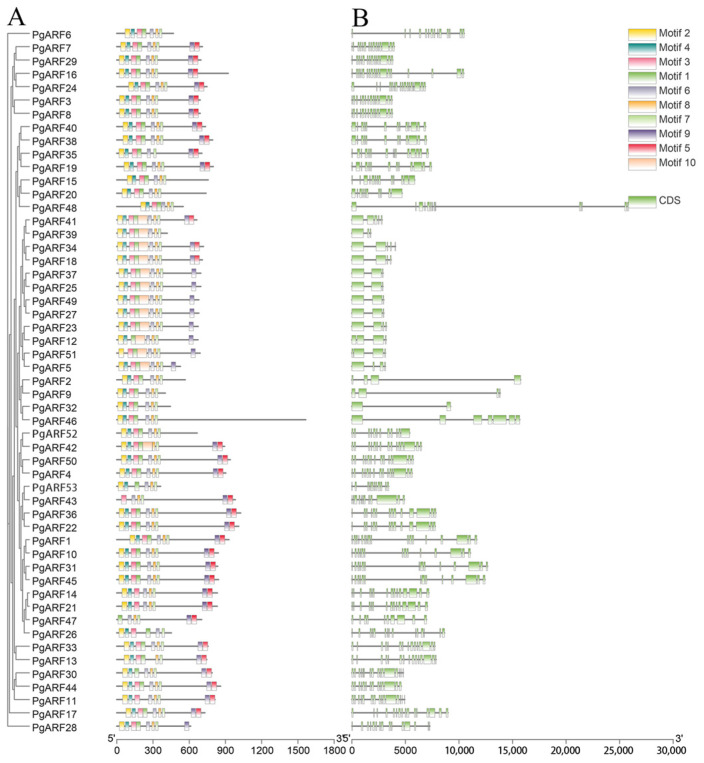
Analysis of protein conserved domains (**A**) and gene structure (**B**) of *PgARF*. Coding and noncoding regions are represented by green boxes and thin lines, respectively; gray lines represent non-conserved sequences, and each conserved domain motif is represented by a numbered colored square to the right.

**Figure 3 plants-12-03943-f003:**
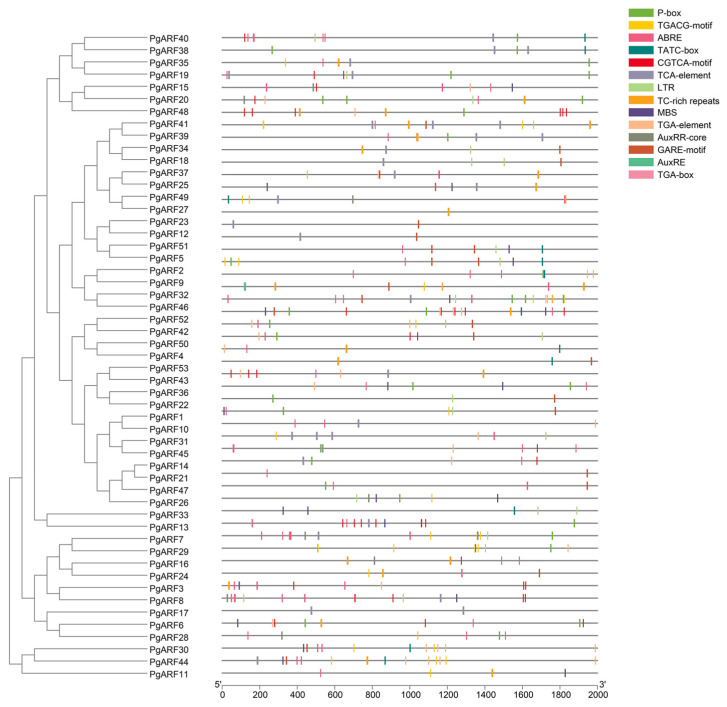
Hormone-responsive and stress-responsive cis-acting elements in ginseng ARF genes. Gray lines represent gene sequences, and each cis-acting element is represented by an annotated colored square on the right.

**Figure 4 plants-12-03943-f004:**
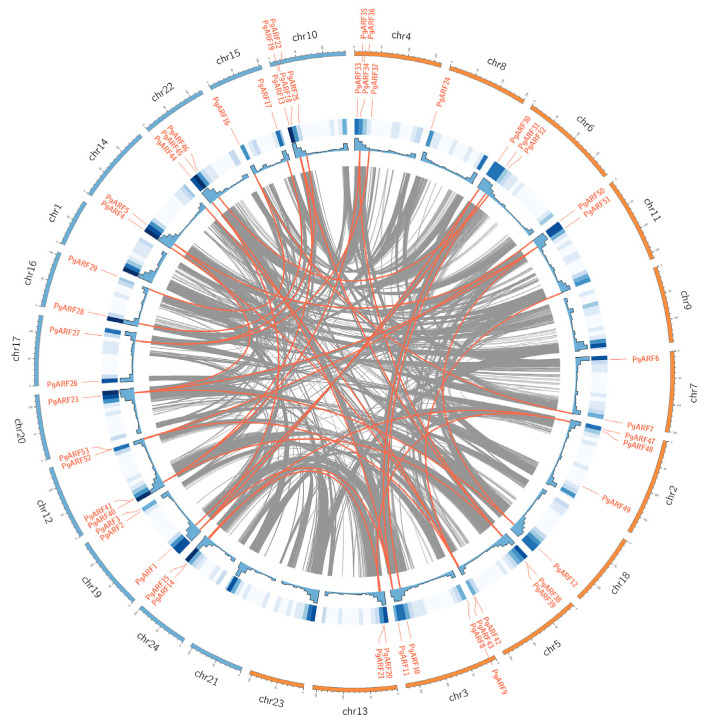
Chromosomal distribution and synteny analysis of ginseng ARF gene. Gray lines indicate all commonality blocks in the ginseng genome, and orange lines indicate duplicated ARF gene pairs. The blue boxes and histograms indicate the gene density of different chromosomes. Chromosome numbers are shown at the top of each chromosome.

**Figure 5 plants-12-03943-f005:**
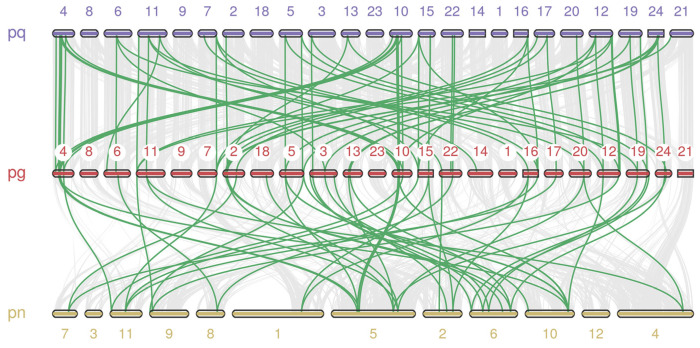
Homology analysis of *Panax quinquefolius*, *Panax notoginseng*, and *Panax ginseng ARF* genes. Gray lines in the background indicate collinear blocks in ginseng and other plant genomes, while green lines highlight collinear gene pairs.

**Figure 6 plants-12-03943-f006:**
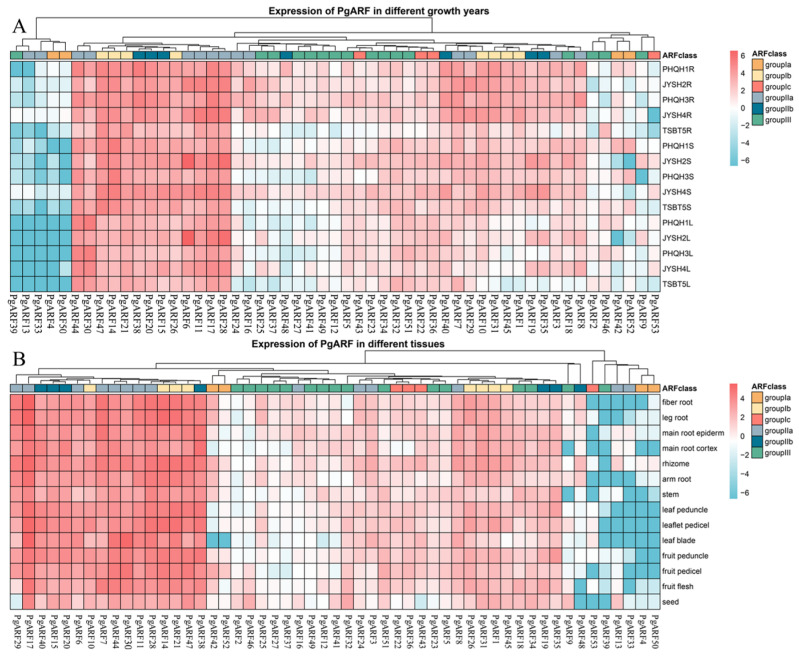
Expression patterns of *PgARF* genes in different growth years and different tissues. (**A**) Expression of *PgARF* genes in different growth years: R, root; S, stem; L, leaf. (**B**) Expression of *PgARF* genes expression in different tissues. The expression level of *PgARF* is represented by red to blue from high to low, and the red and blue on the right indicate the level of expression. The top of the figure shows the grouping of the phylogenetic tree, and the right side shows the color of each subgroup.

**Figure 7 plants-12-03943-f007:**
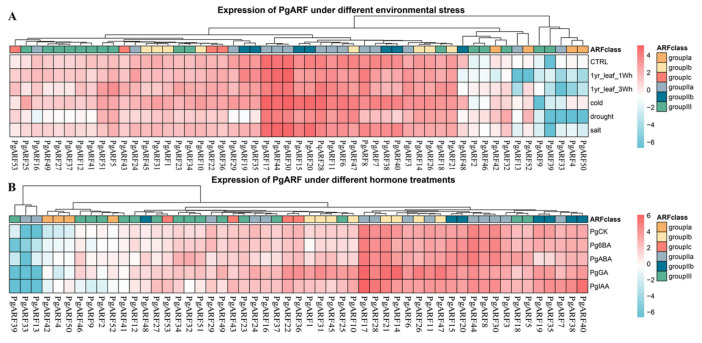
Expression patterns of *PgARF* genes under different environmental stresses and hormone treatments. (**A**) Expression of *PgARF* genes under different environmental stresses; (**B**) expression of *PgARF* genes expression under different hormone treatments. The expression level of *PgARFs* are represented by red to blue from high to low, and the red and blue on the right indicate the level of expression. The top of the figure shows the grouping of the phylogenetic tree, and the right side shows the color of each subgroup. CK–control group; 6BA—cytokinin; ABA—abscisic acid; GA—gibberellin; IAA—auxin.

**Figure 8 plants-12-03943-f008:**
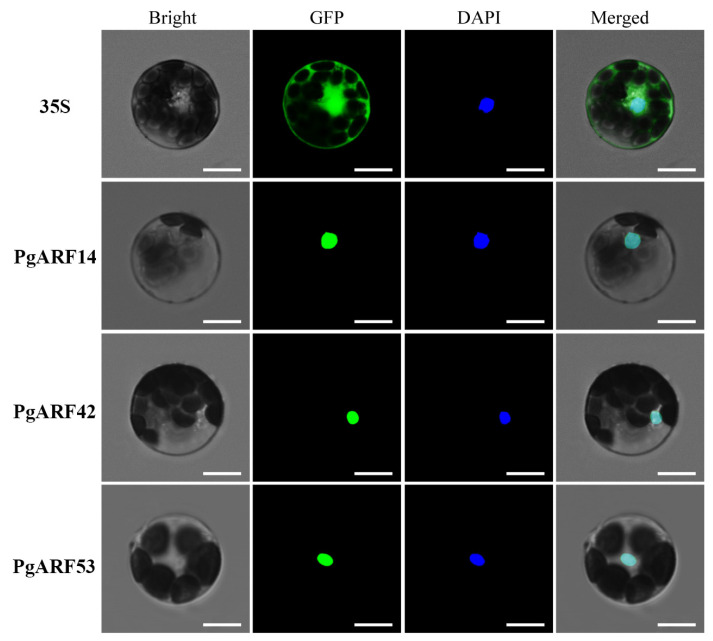
Subcellular localization analysis of *PgARF14*, *PgARF42*, and *PgARF53*. Scale bar = 10 μm.

**Figure 9 plants-12-03943-f009:**
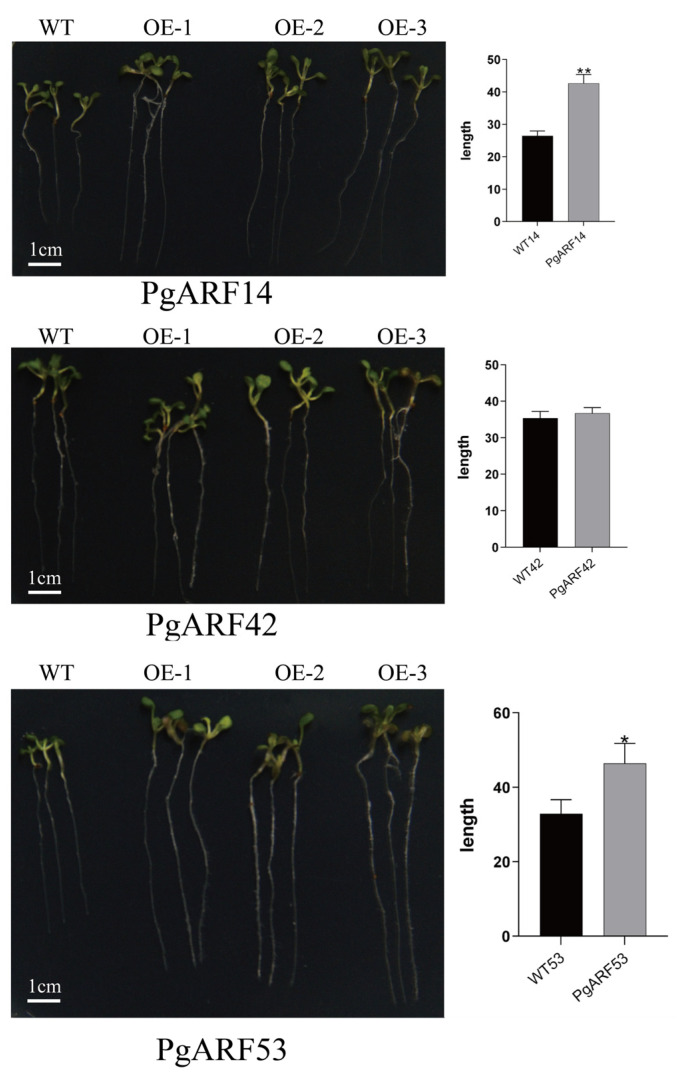
Overexpression of *PgARF14* and *PgARF53* in *Arabidopsis* promotes root development. Significant analysis of root development morphology and corresponding main root length difference between *PgARF14*, *PgARF42*, and *PgARF53* transgenic *Arabidopsis* and wild control group. The first group from the left is the control group, and the three groups to the right are three groups of transgenic *Arabidopsis* for each gene. * is significant difference, ** is extremely significant difference.

## Data Availability

Data is contained within the article and Appendix A.

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
