# Peer review of "Genome-Wide Identification and Expression Analysis of Auxin Response Factor (ARF) Gene Family in Panax ginseng Indicates Its Possible Roles in Root Development"

_plants, 2023, doi:10.3390/plants12233943_

Round 1

Reviewer 1 Report

Comments and Suggestions for Authors

The study deal with the genome-wide identification and expression analysis of auxin response factor (arf) gene family in Panax ginseng highlighting its role in root development. The study provides a theoretical and experimental framework for investigating role of hormones in plant growth. However, there are some limitations which must be addressed.

Specific methods and results should be provided in the abstract, such as number of introns, and exons, highest numbers etc.

Line 59 “numerous ways” specify or name at least the most common.

Line 91 “Panax ginseng’ species name must be italic everywhere in the MS.

Line 37-38 should be cited with recent studies. The following studies could be included. doi: 10.1038/s41467-022-32364-3,https://doi.org/10.1016/j.xplc.2023.100597, https://doi.org/10.1016/j.plaphy.2021.01.042,

Line 45 “Auxin response factor (ARF)” is already explained in the abstract, subsequently use abbreviation. Similarly, in case of others.

Last paragraph of the study should discuss novelty of the study.

Figure 1 legend must have description of the clades I, II ..

Figure 2 would be better to include a clean and readable version.

The gene names should be italic in the whole study.

Line 346 in discussion should provide some more studies references. The following studies could be cited.  https://doi.org/10.1016/j.plaphy.2021.01.042, https://doi.org/10.3390/ijms22179175

 Line 419-420 which previous analysis?

Line 420-423 add the study of Xiong et al, 2023, discuss and compare with the findings. https://doi.org/10.1016/j.xplc.2023.100597

In discussion section most of the sentences are not cited. It seems that the study is not compared with related studies. Discussion should be improved in the light of related studies.

“MCScanX software” provide the link

4.6 details about moisture contents of the green house and soil conditions should be provided.

Conclusion of the study is missing, also provide recommendations and future prospects of this study.

Comments on the Quality of English Language

Typos and grammar should be check also italicize plant names and gene names. 

Reviewer 2 Report

Comments and Suggestions for Authors

The manuscript by Yan et al reported the bioinformatics analysis of the ginseng ARF gene family and preliminary analysis of three PgARFs. The authors did some stylized analyses on the PgARF genes, but some conclusion is not supported by their results. The most worthwhile aspects of this work, in my view, are the transgenic analysis of three PgARFs. However, it is so superficial that we could not draw conclusions from the experimental data in the manuscript. The figure is the worst one among my reviewed papers, and the resolution could not meet the requirement of any academic journals including PLANTS. In addition, the manuscript is not well written, and there are a few grammatical mistakes and errors.

Major problems:
1) Please clarify the reason for only choosing PgARF14/42/43 for transgenic analysis. In Line 246-298, the author stated that there so many PgARFs associated with developmental cure as well as phytohormone inductions.
2) Analysis of transgenic lines: which generation of transgenic seedlings were analyzed? Did you check the expression levels of transgene by qRT-PCR method? As PgARF53 exhibited such lower level of expression in root tissues, how can you conclude its contributions to root growth according to your ox lines?  

Minor problems:
1) Please check the whole manuscript, use italic for gene and plant Latin name!
2) In Fig 6, ARF3c??.
3) Line318-320, “PgARF14 was markedly longer than….”, the primary root length of PgARF53” ?? Please rewrite the sentences.

Reviewer 3 Report

Comments and Suggestions for Authors

This work is well presented and English is good. The technology adopted very up to date . The results obtained confirm previous results on  the ARF mode of action

Author Response

We sincerely appreciate your  constructive comments on our work. Moving forward, we will do more comprehensive research to further explore the functions of ginseng ARF genes.

Round 2

Reviewer 1 Report

Comments and Suggestions for Authors

All comments are addressed and the study can be accepted for publication

Author Response

Thanks for your kindly reply.

Reviewer 2 Report

Comments and Suggestions for Authors

It seems that the authors did not understand my concerns in the comment 2. Please read it carefully, and fix the manuscript.

Round 3

Reviewer 2 Report

Comments and Suggestions for Authors

OKay.